# Genome Size Variation Is Associated with Hybrid Vigor in Near-Isogenic Backgrounds in *Brassica napus*

**DOI:** 10.3390/plants14193013

**Published:** 2025-09-29

**Authors:** Rui Wang, Meicui Yang, Haoran Shi, Yun Li, Jin Yang, Wanzhuo Gong, Qiong Zou, Lanrong Tao, Qiaobo Wu, Qin Yu, Hailan Liu, Shaohong Fu

**Affiliations:** 1Maize Research Institute of Sichuan Agricultural University, Chengdu 611130, China; 15733875302@163.com; 2The Maize Research Institute at Sichuan Agricultural University’s Chengdu Campus, Chengdu Academy of Agriculture and Forestry Sciences, Chengdu 611130, China; amy2533684145@163.com (M.Y.); 18697480039@163.com (H.S.); liyunxpzh@163.com (Y.L.); yjjing@163.com (J.Y.); gongwanzhuo@hotmail.com (W.G.); zou_qiong123@163.com (Q.Z.); tlr325@163.com (L.T.); wuqiaobo111@163.com (Q.W.); yu6qin4@163.com (Q.Y.); 3Chengdu Research Branch, National Rapeseed Genetic Improvement Center, Chengdu 611130, China

**Keywords:** double haploid, genome size, heterosis, SNP site, genetic distance

## Abstract

Although heterosis plays a crucial role in enhancing crop yield and stress resistance, its underlying genetic mechanism remains not yet fully understood. Previous studies have shown that heterosis tends to increase with greater genetic distance in the absence of reproductive isolation barriers. However, whether variation in parental genome size alone can generate heterosis under near-isogenic backgrounds has not been thoroughly explored. Here, we used a rapeseed double haploid (DH) inducer line to generate progeny from the *Pol* CMS three-line hybrid Rongyou 18 (RY18). Although the progeny maintained the same ploidy level as the parents, their genome sizes showed notable variation (818.99–1024.88 Mb). To eliminate genetic distance effects, multiple DH progeny carrying restorer genes were crossed as paternal parents with the female parent 0068A of RY18, creating novel F_1_ hybrids. Using RY18 as the control, we observed a marked reduction in the genetic distance between the newly induced restorer line and the female parent (0068A). Correlation analysis further revealed a significant negative correlation (r = −0.310 *) between the paternal genome size and heterosis for thousand-seed weight (TSW). Furthermore, the genomic expansion in hybrid offspring relative to the male parent showed that significant correlations were observed between paternal genome size and heterosis over the standard for both TSW (r = 0.300, *p* < 0.05) and plot yield (r = 0.326, *p* < 0.05). Resequencing of high-and low-yielding F_1_ hybrids identified SNP sites, indicating that under an identical genetic background, heterosis for yield was more pronounced on chromosome A and chromosome C04. The doubled haploid (DH) induction line facilitates the generation of parental lines with distinct genome sizes, potentially providing a potential novel approach for studying heterosis research in *Brassica napus*.

## 1. Introduction

Plant genome size, typically expressed as the C-value, refers to the total DNA content of a genome and serves as a critical reference for evaluating and utilizing germplasm resources [1]. Studies have revealed extensive variation in eukaryotic genome sizes, with differences exceeding 64,000-fold [2]. The observed diversity in plant genome sizes is largely attributed to polyploidy and the presence of polyploidy and repetitive sequences [2,3,4]. Nearly all plants have undergone one or more ancestral polyploidy events, which have notably impacted genome size and allele content [5]. The extensive accumulation of repetitive sequences, particularly long terminal repeat retrotransposons (LTR-RTs), constitutes the most abundant component of genomic DNA and represents a major driver of genome expansion [6]. Both polyploidy and LTR insertions not only increase genome size and organismal complexity in some species but also generate novel functions, alter gene expression patterns, and enhance plant survival under harsh environmental conditions, thereby contributing to the evolutionary fitness of certain lineages to the present day [6,7,8]. Significant genome size variation exists within *Brassica napus*. Exploiting parental genome size differences, especially those of the female parent, carries strong practical significance for improving rapeseed breeding germplasm and generating high-heterosis hybrid combinations [9].

Heterosis refers to the phenomenon in which hybrid offspring (heterozygotes) derived from genetically distinct parents exhibit enhanced growth vigor, improved stress tolerance, and greater adaptability compared with both parental lines [10]. This effect is widely exploited in crop breeding and has been observed in rapeseed, rice, maize, sorghum, and tomatoes [11,12,13,14]. In rapeseed, the utilization of heterosis is marked by pronounced phenotypic effects, high and stable yields, and relatively low seed production costs [15]. The traits associated with heterosis are primarily quantitative, controlled by numerous minor-effect polygenes. Nevertheless, the precise molecular mechanisms underlying heterosis remain incompletely understood. Current research, employing molecular markers and QTL mapping, extensively supports and investigates three major hypotheses: dominance, overdominance (superdominance), and epistasis [16]. Although molecular evidence exists for all three hypotheses, none is regarded as the sole explanatory mechanism [17]. Studies suggest that differences in DNA content between hybrids and their parental lines may contribute to heterosis. Rayburn measured nuclear DNA content in several maize (Zea mays) inbred lines and their F_1_ hybrids, finding that some hybrids exhibited DNA content equal to or significantly exceeding the parental median [18]. This finding indicated instability in hybrid nuclear DNA content and a corresponding variability in the amount of heterosis. Similarly, Liu observed higher relative nuclear DNA content in rice hybrids compared to parents, suggesting a potential link between increased DNA content and heterosis in rice [19]. Zhu, investigating the effects of genome size on heterosis in rapeseed, reported significant positive correlations between parental genome size and F_1_ plant height, branch height, and single-plant yield, along with a highly pronounced negative correlation with protein content [9]. Based on these findings, it was inferred that hybridization may promote genome expansion in the F_1_ generation, thereby contributing to heterosis. Furthermore, differences in parental genome size were found to be significantly associated with heterosis for F_1_ traits such as plant height, thousand-seed weight, and protein content.

Fu identified the rapeseed DH induction line Y3380 (AAAACCCC, 2n = 8x = 76), an artificially synthesized heterologous octaploid *Brassica napus* L. [20]. When used as a male parent to pollinate other rapeseed of female lines, Y3380 can directly induce maternal DH plants, with an induction rate ranging from 30% to 90%. The paternal chromosome elimination induced by the haploid inducer triggers the production of double haploids in *Brassica napus* [21]. Another study found that a rapeseed DH-induction line was able to induce the homozygous maternal plants of the nap cytoplasmic male sterile line, while simultaneously enabling the selection and breeding of both the sterile and maintainer line [22]. The induction of rapeseed DH lines is characterized by a substantial shortening of the breeding cycle and the rapid generation of stable, homozygous lines, playing a critical role in the development of the three-line system. In the present study, a rapeseed DH induction line was employed to further investigate the relationship between genome size and yield heterosis within the same genetic background, and analyze the chromosomal distribution of SNPs associated with high heterosis on chromosomes, aiming to provide a novel perspective for the utilization of heterosis in rapeseed.

## 2. Results

### 2.1. Fertility and Restore Genetic Markers Identification of Induced Offspring

To select the induced offspring plants with the *pol* sterility recovery gene, 89 induced offspring lines (F_2_) were analyzed. The *pol* CMS restorer line C2970 served as the positive control, while the sterile line 0068A served as the negative control. Among the induction lines, Y3380 lacked the marker, whereas RY18 carried it. Molecular detection identified 56 lines as positive (Figure 1), indicating that the induced line Y3380 does not possess the *pol* recovery gene, while RY18 and 56 induced offspring do. Fertility assessment of the RY18-induced offspring confirmed that all 56 lines were fertile (Table A1), consistent with the molecular results. Fertile plant 157-9 (Figure 1A–D) exhibited slightly larger petals, the stamens with filaments exceeding the stigma, and abundant, active pollen. In contrast, sterile plants 157-19 (Figure 1E–H) showed smaller petals, atrophied stamens, there is no pollen or only a small amount, little or no pollen, and low or absent pollen viability. Overall, the fertility assessment results closely matched the molecular marker identification.

### 2.2. Ploidy and Chromosome Identification Number of Induced Offspring

Further ploidy analysis is required for the 56 previously screened induced offspring to confirm the success of the induction. Three offspring with different genome sizes and one offspring with an abnormal genome size were randomly selected for cytological observation and ploidy determination. According to flow cytometry identification, the fluorescence signal in the G1 phase of 157-16, 3689-4, 3689-14, and 3689-26 are 333,633.47, 547,692.81, 651,086.08, and 692,118.19, respectively (Figure 2A–D; Table A2). The chromosome numbers of 157-16, 3689-14, and 3689-4 were 38, with no observable chromosomal abnormalities, and these plants were identified as tetraploids. The chromosome number of 3689-26 was approximately 52, exhibited chromosomal lag, and was identified as hexaploid (Figure 2E–H). These results are consistent with the flow cytometry analyses.

### 2.3. SNP Chip Identification and Analysis

Based on the identification of the *pol* CMS restorer gene, SNP chip analysis was conducted on induced offspring carrying the restorer gene together with parental lines RY18, C2970, 0068A, and Y3380. Genetic distance analysis using these SNP data revealed that the greatest distance occurred between RY18 and Y3380 (0.537), while the distance between C2970 and 0068A was 0.357. All induced restorer lines showed genetic distances <0.357 when compared with 0068A. Among the parental lines, induced offspring exhibited the closest genetic relationship to RY18 (smallest average distance) and the most distant relationship to Y3380 (largest average distance). Furthermore, the genetic distances between induced offspring and both 0068A and C2970 were smaller than those between RY18 and these same parents, suggesting that induction reduced the genetic divergence between RY18-derived offspring and their parental lines. Overall, induced offspring were genetically closer to 0068A, C2970, and RY18 but more distant from Y3380 (Table 1 and Table A3).

The genetic relationship clustering diagram (Figure 3) reveals two major clades. Y3380 and 3689-26 cluster together within the same first clade, consistent with their inferred origin as hybrid offspring derived from crossing an induced line with RY18, which also agrees with cytological observations. All other germplasm falls into the second major clade, within which the induced offspring are distributed across four sub-branches: C1, C2, D1, and D2. Parental lines RY18 and 0068A co-localize within the D branch, whereas C2970 groups within the C branch. Notably, the D branch contains more induced offspring than the C branch. This clustering pattern indicates that progeny generated by pollinating Y3380 with RY18 pollen were primarily the result of induced hybridization events.

### 2.4. Flow Cytometry Detection and Analysis

Flow cytometry was employed to measure G1-phase fluorescence intensity in the induced offspring lines, with ZS11 (reference genome size: 966.00 Mb; Table A4) as the internal standard. The relative standard deviation (RSD) of G1-phase fluorescence intensity across lines ranged from 0.59% to 8.06%, indicating high measurement stability. Genome size showed notable variation among the induced lines: the smallest was 818.99 Mb in line 4543-4, representing a 15.22% decrease relative to ZS11, while the largest was 1024.48 Mb in line 4548-1, representing a 6.05% increase. The smallest genome was 818.99 Mb in line 4543-4, representing a −15.22% decrease relative to ZS11, while the largest was 1024.48 Mb in line 4548-1, representing a 6.05% increase. The overall difference between the maximum and minimum values was 25.09%. Taken together with cytological observations, these findings demonstrate substantial genome size variation among induced offspring lines under homoploid conditions.

To assess whether genome size variation occurs in conventionally bred progeny, we compared the F_2_ generation derived from conventional self-pollination (RY18-F_2_ and RY18-F_3_) with induced offspring lines. Kolmogorov–Smirnov (K-S) tests confirmed that genome sizes in all three populations—induced offspring, RY18-F_2_, and RY18-F_3_—followed normal distributions (Table 2; Figure 4).

### 2.5. Investigation Results of Agronomic Traits

An innovative restorer line (induced from RY18 via Y3380 pollination) was crossed with the male-sterile line 0068A to produce hybrid F_1_ progeny, with RY18-derived hybrids serving as controls (CK). Results showed that the average values of the eight yield traits, plant height, root and stem thickness, main raceme fruiting segment length, number of effective primary branches, number of effective siliques on the main axis, number of effective siliques per plant, number of seeds per plant and theoretical yield per plant, as well as the quality trait of oil content, were all higher than those of RY18 (CK), and exhibited positive average heterosis values. In contrast, the average values for branch height, thousand-seed weight, and actual plot yield, as well as the quality traits of protein content, glucosinolate content, and erucic acid content, were all lower than those of RY18 (CK), with negative average heterosis values (Table A5 and Table A6). Notably, the highest over-standard heterosis was observed for theoretical yield per plant, reaching 91.01%.

### 2.6. Resequencing Analysis

Resequencing was conducted on seven high- and low-yielding hybrid combinations, their corresponding parental lines, and RY18 with its parental lines. Genetic distances between seven new F_1_ hybrids (Table A7) and their male parent, RY18, and their male parent, C2970, as well as the female parent, 0068A were calculated based on SNP sites, and clustering plots were generated (Figure 5). The genetic distance between C2970 and 0068A is 0.4875, representing the most distant relationship, which was consistent with the SNP chip clustering results. The closest genetic relationship was observed between the new hybrid F_1_-13 and the female parent 0068A, with a genetic distance of 0.1041. The genetic distances of RY18 from 0068A and C2970 were 0.2663 and 0.2812, respectively. On average, the genetic distances of the new F_1_ hybrids from 0068A and their corresponding male parents were 0.1601 and 0.2119, respectively. Both RY18 and the new F_1_ hybrids showed smaller genetic distances from the female parent, indicating a stronger genetic contribution from the female parent. Furthermore, the average genetic distance between the new F_1_ hybrids and RY18 was 0.1685, smaller than the distance from their male parent (0.2119). These findings suggest that under the same genetic background, the shared female parent results in a closer genetic relationship between the new hybrid F_1_ hybrids and RY18 (Table 3 and Table A8).

SNP sites were characterized for seven novel F_1_ hybrids (Table A9), yielding 332 high-yield-associated and 1170 low-yield-associated SNP sites. After excluding sites shared between yield groups, chromosomal distribution analysis revealed that 138 high-yield-associated and 892 low-yield-associated SNPs were located on chromosome A, and 79 high-yield-associated and 163 low-yield-associated SNPs were located on chromosome C. This distribution indicates that both high- and low-yield-associated SNPs are predominantly enriched on chromosome A, which also shows a stronger correlation with yield variation compared to chromosome C. Notably, SNPs associated with both high and low yield are significantly clustered on chromosomes A06, A09, C03, and C04, indicating that these specific chromosomal regions are tightly linked to yield traits.

### 2.7. Correlation Analysis of Traits

Hybrid offspring were generated by crossing the sterile line 0068A (female parent) with multiple distinct male parents. The offspring shared the same female parent but had different male parents. Correlation analysis between genome size and genetic distance showed that only the genome size of the male parent is significantly correlated with the genetic distance between the progeny and the male parent, with a correlation coefficient of 0.790* (Table 4). Further correlation analysis of genome size, male parent genetic distance, and the superiority of agronomic traits (Table 5) revealed that the male parent’s genome size was significantly negatively correlated with the superiority of thousand-seed weight (–0.310), but positively correlated with the superiority of thousand-seed weight increase (0.300 *) and actual plot yield (0.326 *). These results indicate a significant correlation between the size of the male parent genome and heterosis. Parental genetic distance was significantly correlated only with the heterosis of thousand-seed weight and root and stem thickness. Specifically, the correlation coefficient with the heterosis of root and stem thickness was 0.346 *, indicating a significant positive relationship, whereas the correlation coefficient with the heterosis of thousand-seed weight was −0.325 *, indicating a significant negative correlation. These results suggest a notable association between the male parent’s genome size and overall heterosis. In contrast, parental genetic distances were significantly correlated with heterosis for only a few traits in this study.

## 3. Discussion

Haploid induction (HI) was first demonstrated in maize in 1958, and HI systems have since been developed for numerous crops [23,24,25,26,27,28]. Although haploids generally require artificial chromosome doubling due to low spontaneous doubling rates, rapeseed possesses a unique doubled haploid (DH) induction line [20]. This line serves as the male parent in crosses, enabling the direct production of DH progenies. The dominant rapeseed hybrid RY18, derived from a DH induction line, can generate homozygous offspring within just two generations. The “C-value paradox” highlights the absence of a consistent relationship between DNA content and organismal complexity [29]. Further investigations have revealed significant, non-random genome size variation within taxonomic groups and species [30]. For example, Walker measured the genome size of six *Bituminaria bituminosa* populations from three distinct locations, and attributed up to 11% variation to geographic factors [31]. However, other studies have indicated that genome size variation may be spatially random and not correlated with environmental conditions, with differentiation initially lacking adaptive significance [32]. In the present study, all experimental materials were sourced from the same environment and experimental field, and leaf samples were collected from identical positions on each plant. This design minimizes potential influences from factors such as climate, altitude, soil conditions, and sampling location. Genome sizes were measured in the hybrid RY18 (induced via the DH line) and its parental lines. Importantly, all induced progenies were confirmed to be tetraploid. Results revealed up to 20% genome size variation within rapeseed (*Brassica napus*), indicating substantial genome size differences exist not only between ploidy levels but also among conspecific populations sharing identical ploidy and origin.

Long terminal repeat retrotransposons (LTR-RTs) represent the predominant class of repetitive sequences in the lily genome. A massive and irreversible burst of LTR-RT insertions over a short evolutionary period has driven the rapid expansion of the lily genome [33]. However, genome expansion is not limitless: transposon-mediated homologous unequal recombination and irregular recombination events can remove DNA sequences, thereby constraining overall genome size. Among transposable elements, retrotransposons are the most abundant DNA components, playing key roles in determining genome size variation, shaping plant architecture, contributing to intraspecific diversity, altering genome structure, and regulating gene expression to modulate phenotypic traits [34]. Thus, exploring the relationships among genome size variation, phenotypic traits, and heterosis may provide insights into the functional role of retrotransposons. Previous studies have shown that the genomic size of hybrid offspring can differ from that of their parents, which may partly account for the occurrence of heterosis. In this study, 15 phenotypic traits were evaluated, and correlations were analyzed among genomic size, parental genetic distance, and trait performance. Results indicated that only the thousand-seed weight and single-plant yield were significantly correlated with genomic size, indicating a strong association between genomic size and yield-related heterosis. By contrast, parental genetic distance was significantly correlated only with root and stem thickness and thousand-seed weight. Given the largely uniform genetic background of the materials and the relatively small parental genetic distance, its relationship with heterosis could not be effectively demonstrated. Therefore, while parental genetic distance may serve as a reference in practice, its predictive value for heterosis remains uncertain in this context.

SNPs represent DNA sequence polymorphism caused by variations at a single nucleotide site, including both transitions and transversions. The number of seeds per silique is a key yield component in rapeseed, and elucidating its genetic basis is critical for improving yield potential. In recent years, numerous Quantitative Trait Loci (QTLs) for silique length (SL) and the number of seeds per silique (NSPS) have been identified across nearly all chromosomes through QTL mapping and genome-wide association study (GWAS). For example, Sang utilized a 60K SNP array to genotype 14 parental lines, estimated genetic distance based on SNP data, and demonstrated that genetic distance could partially predict heterosis for single-plant yield, branch height, and plant height [35]. Similarly, Sun conducted GWAS on an association panel of 496 representative rapeseed accessions and detected 20 significant loci distributed on chromosomes A01, A04, A07, A08, C01-C07, and C09, of which six overlapped with previously reported QTLs. Several candidate genes were also identified, providing a foundation for the genetic improvement of NSPS in rapeseed [36]. More recently, Liu performed GWAS on NSPS in *Brassica napus* and identified a novel gene *BnOFP13_2*, significantly associated with this trait, highlighting its potential application in rapeseed breeding [37]. This study resequenced 17 samples to analyze the number and chromosomal distribution of SNPs associated with high and low yields The results showed that both high-yield and low-yield- associated SNPs were predominantly located on the A subgenome, Specifically, high-yield-associated SNPs were enriched on A06 and C04, whereas low-yield-associated SNPs were enriched on A01, A03, A09 and C03, C04, C06, C08 (relative to other chromosomes within their respective subgenomes). This distribution pattern is consistent with the core SNPs identified by Wang [38]. Overall, SNPs tended to cluster on chromosome A, with both high- and low-yield-associated SNPs concentrated on C04, and low-yield-associated SNPs additionally enriched on A03. These findings suggest that, under conditions of reduced genetic distance, heterosis may be driven by the interaction of a relatively small number of loci. Conversely, when genetic distance is minimized, the accumulation of additional SNPs appears to reduce heterosis, indicating that heterosis-associated loci are not uniformly distributed across the genome.

## 4. Experimental Materials and Methods

### 4.1. Experimental Materials

All the experimental materials of this study were provided by the Chengdu Academy of Agriculture and Forestry Sciences. Y3380 is an artificially synthesized rapeseed DH-induced line, which is octaploid (2n = 8x ≈ 76, AAAACCCC). All the following materials are tetraploids (2n = 4x = 38, AACC). 0068A is a polycytoplasmic sterile line, and C2970 is a polycytoplasmic sterile restorer line. Both are double-low Brassica napus. Rongyou 18 (RY18) is a double-low, high-yield, high-oil, and multi-resistant three-line hybrid rapeseed variety formed by combining the 0068A sterile line and the C2970 restorer line. Lines 3689, 157, 4537 to 4558 are the induced offspring lines (F_2_) obtained by inducing RY18 with Y3380 as the paternal parent. 3688 is the self-crossbred F_2_ generation of RY18 (RY18-F_2_), and 4529 to 4536 are the self-crossbred F3 generation of RY18 (RY18-F3). All plant materials were grown at the Chongzhou Yangma Base of Chengdu Academy of Agricultural and Forestry Sciences. The experimental workflow diagram is shown in Figure 6.

### 4.2. DNA Extraction and Detection

The DNA of the induced offspring was extracted by the Novozyme kit method. The concentration and quality of DNA were detected by a nucleic acid protein detector. The concentration was above 50 ng/μL, the absorbance ratio of 260/280 was between 1.8 and 2.0, and the ratio of 260/230 was above 1.8. The test was conducted using 1% agarose gel electrophoresis. Samples with a single bright band greater than 2000 bp were selected, then diluted to approximately 100 ng/μL and stored at −20 °C for future use.

### 4.3. Molecular Marker Identification Restores Genes

Restore the genetic markers for the induced offspring (Table 6). PCR amplification for restorer gene identification was performed using the following thermal cycling conditions: initial denaturation at 94 °C for 3 min; 30 cycles of denaturation at 94 °C for 30 s, annealing at 57 °C for 30 s, and extension at 72 °C for 1 min; followed by a final extension at 72 °C for 5 min. The amplified fragments were isolated and visualized using a 2.5% agarose gel with ethidium bromide staining. After electrophoresis, they were observed, photographed, and analyzed in a gel imaging system.

### 4.4. SNP Purity Identification

The first step of this experiment involved extracting DNA from leaf samples. The extracted DNA was then diluted to a concentration range of 50–100 ng/μL. Sample detection: The next set of procedures included whole genome amplification, DNA fragmentation, DNA purification, DNA resuspension, DNA denaturation, DNA hybridization with a chip, single base extension, and staining. Once these steps were completed, the chip was scanned, and the data were typed. The chip detection was performed by Wuhan Shuanglvyuan Biotechnology Company. The 50K SNP chip is in uniform coverage of 19 pairs of chromosomes in *Brassica napus* developed by Wuhan Shuanglvyuan Biotechnology Company, which is based on the Illumina Infinium SNP chip technology. This chip was used for detection, typing analysis, and initial positioning.

### 4.5. Ploidy Detection by Flow Cytometry

The fresh young leaves were collected, cleaned with distilled water, and dried. A small round leaf with a diameter of 5 mm was then punched out and placed in a clean and dry glass dish. Next, 0.5 mL of precooled cell lysate was added, and the leaf was quickly shredded with a sharp blade. The resulting mixture was filtered into a 2 mL EP tube using a 35 mm filter head. To this filtrate, 1.5 mL of PI dye solution was added and gently shaken to mix. The tube was then treated in the dark for 30 min. The samples were analyzed using flow cytometry (AccuriC6 Plus model) to detect the fluorescence intensity and the number of cells. Approximately 10,000 cells were collected for each sample. Before detection, ZS11 (a tetraploid Brassica napus, 2n = 38, the genome size is 966 Mb) was used as a control for ploidy detection.

### 4.6. Fertility Identification

Based on the results of molecular marker identification, field identification was conducted on the fertility of induced offspring with and without restored gene markers. Identification criteria: The petals of fertile plants are large, and the stamens develop normally, with obvious pollen visible. The petals of sterile plants are small, the stamens are withered, and there is no pollen scattered. The flowers of several representative fertile and sterile plants were respectively selected and brought back to the laboratory for photography. The pollen was stained with acetic acid and magenta staining solution, and then observed and photographed under a microscope.

### 4.7. Cytological Observation

Some of the induced offspring were selected for cytological observation. We selected the flower buds with a diameter of 2–3 mm, removed the ovary, and placed them in 0.002M 8-hydroxyquinoline for dark treatment at 4 °C for 3–4 h. Then, they were fixed in freshly prepared Carnoy’s fixative (ethanol: glacial acetic acid 16 = 3:1) for 24 h. When handling the anthers, they were directly fixed for 24 h and then stored in a refrigerator at 4 °C with 70% alcohol. When preparing the preparation, we washed the material with distilled water, then placed it in a 1M HCl water bath at 60 °C (6–8 min for the ovary and 2 min for the anther), and washed it again with distilled water after that. After cleaning, the ovary or anthers were placed on a slide and crushed to remove the residue. Then, a modified phenol magenta staining solution was added, and a cover slip was used to gently tap and press the slide. After the slide was made, the chromosome behavior was observed, the chromosome number was counted, the ploidy was identified, and photos were taken under a 100x oil immersion optical microscope.

### 4.8. Investigation of Agronomic Traits

The 54 newly harvested hybrid F_1_, control RY18, and their parents were sown into the Chongzhou Yangma Base of Chengdu Academy of Agricultural and Forestry Sciences. Each F_1_ type has 10 lines, with 3 repetitions set; each RY18 type has 10 lines and set 6 repetitions. We induced each of the five lines of 0068B (0068A maintainer line) and C2970 of the offspring, without repetition. After the maturity period, 5 plants were randomly selected from 54 F_1_ and RY18 repeated plants for statistical investigation of various agronomic traits. Meanwhile, 30 corner fruits of each F_1_ and RY18 were randomly selected to calculate the average number of grains per corner fruit. After drying, their thousand-grain weight was weighed. In May, all F_1_ and RY18 are harvested, threshed, impurities removed, and weighed. The erucic acid content, oil content, protein content, and other quality traits of the seeds were determined by a 5000-type near-infrared analyzer. Over-standard heterosis refers to the percentage or absolute difference by which the phenotypic value of a target trait in the first filial generation (F_1_) exceeds that of the corresponding trait in a specific standard control variety. Over-standard heterosis (%) = [(Mean phenotypic value of the trait in F_1_ generation—Mean phenotypic value of the trait in the standard control variety)/Mean phenotypic value of the trait in the standard control variety] × 100%

### 4.9. Resequencing

#### 4.9.1. Sample Sequencing, Sequence Alignment, and Variant Detection

The actual cell yield with the least overproduction advantage of hybrid F_1_ was selected as the sequencing sample trait. Four high-yield F_1_ (F_1_-9, F_1_-34, F_1_-36 and F_1_-41) and three low-yield F_1_ hybrids (F_1_-2, F_1_-3, and F_1_-13) were selected. These seven F_1_ hybrids were then combined with the control RY18 and their respective parent samples. A total of 17 sample leaves were sent to Bio-Tech Biotechnology Co., Ltd. for third-generation whole gene resequencing, with a sequencing depth of 50×. The NGS QC Toolkit v2.3.3 software was used to conduct quality inspection on the sequencing raw data to obtain high-quality data. The clean reads of 17 samples were aligned to the rapeseed reference genome ZS11.fa by BWA, and SNP sites detection was completed using GATK v4.2.6.1 software.

#### 4.9.2. Sample Genetic Distance Cluster Analysis

The genetic distance between each sample was calculated using VCF2Dis-main v1.43 software, and cluster analysis and plotting were carried out using MEGA 11 software.

#### 4.9.3. Analysis of the Differences in SNP Sites Between High and Low Yields

High and low-yield-associated SNPs were identified by comparing allele frequencies between high-yielding (top 25%) and low-yielding (bottom 25%) F_1_ hybrids. Common SNPs present in both groups were excluded to isolate yield-specific variants. The chromosomal distribution of these SNPs was visualized using the CM plot package v3.6.2 in R.

## 5. Conclusions

In conclusion, DH-induced lines not only offer advantages but also address limitations associated with continuous self-pollination, particularly in applications related to genome size. Under the confirmed condition that all induced progenies are tetraploids, genome size variation was nonetheless observed among plants of the same species, ploidy, and origin. The rapeseed DH-induced line can thus generate parental lines with distinct genomic sizes. Significant correlations were detected between paternal genome size, parental genetic distance, and heterosis, highlighting the complexity of heterosis formation. Given that no single approach can accurately predict heterosis, a more comprehensive prediction framework should integrate genome size, genetic distance, and key trait-associated genes. Such an approach has the potential to reduce breeding costs and accelerate the breeding process, thereby providing valuable guidance for future crop improvement. Furthermore, SNP analysis of high- and low-yielding lines suggests that chromosome A is strongly associated with yield, with chromosome A06 and C04 in particular harboring a relatively high number of yield-related SNPs and potential core genes. These findings indicate that C04 may carry more functional loci, underscoring the importance of chromosomes A, A06, and C04 in future studies on the molecular mechanisms underlying yield formation and regulation.

## Figures and Tables

**Figure 1 plants-14-03013-f001:**
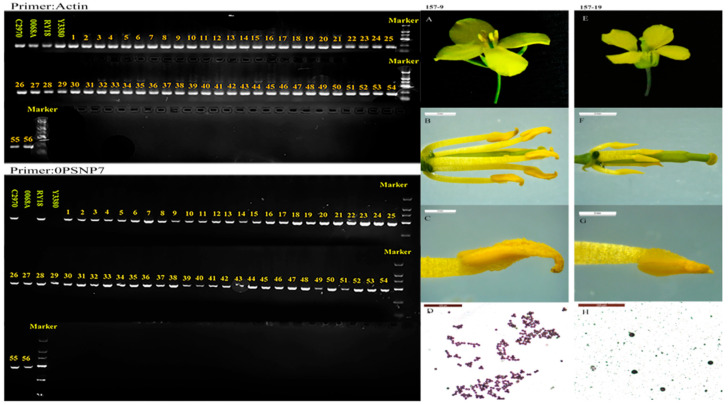
The results of molecular marker identification of *Pol* CMS restorer gene. The comparison of flower morphology between fertile and sterile induced progeny plants. (**A**–**D**) The flower morphology of fertile plants; (**E**–**H**) the flower morphology of sterile plants. The scale bars of (**B**,**C**,**F**,**G**) are 2 mm, and the scale bars of (**D**,**H**) are 100 μm.

**Figure 2 plants-14-03013-f002:**
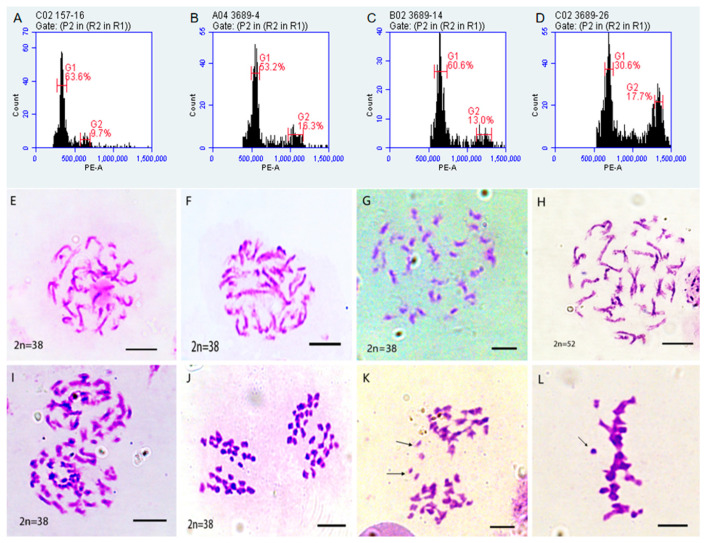
The ploidy identification and chromosome behavior of induced progeny. X axis: Represents the relative intensity of the fluorescence signal. Y axis: Represents the number of cells. (**A**–**D**) The flow histograms of 157-16, 3689-4, 3689-14, 3689-26; (**E**–**H**) the mitosis of 157-16, 3689-4, 3689-14, 3689-26; (**I**,**J**) the meiosis of 157-16 and 3689-14; (**K**,**L**) the meiosis of 3689-26. The arrow in the figure points to a lagging chromosome.

**Figure 3 plants-14-03013-f003:**
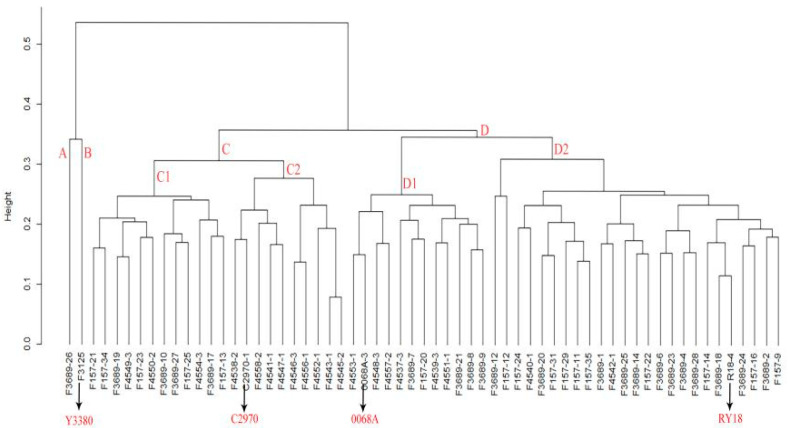
The clustering diagram of genetic distance between induced offspring and Y3380, RY18, and parents of RY18. Y axis: Represents the relative distances of each category.

**Figure 4 plants-14-03013-f004:**
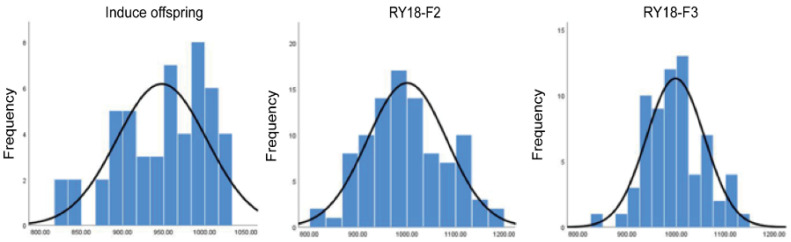
The histogram of genome size of induced offspring and self-crossing progeny by RY18.

**Figure 5 plants-14-03013-f005:**
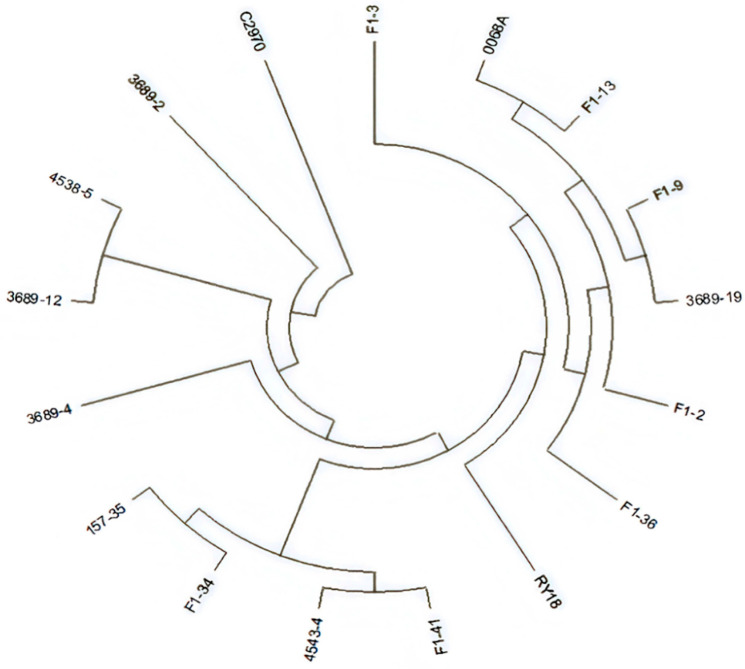
The clustering diagram of genetic relationship.

**Figure 6 plants-14-03013-f006:**
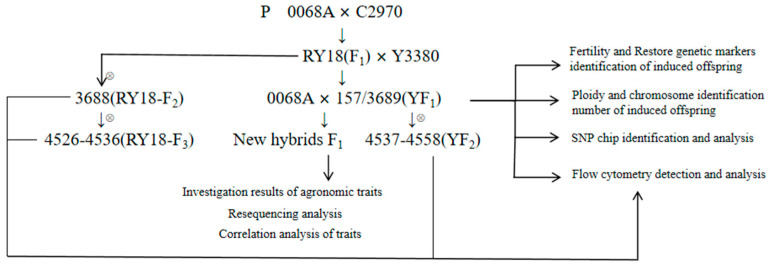
The experimental workflow diagram.

**Table 1 plants-14-03013-t001:** The analysis of genetic distance.

	RY18	Induce Offspring (Mean)	YF_1_ (Mean)	YF_2_ (Mean)
0068A	0.346	0.257	0.264	0.235
C2970	0.336	0.254	0.273	0.227
Y3380	0.537	0.455	0.466	0.435
RY18	0.000	0.201	0.174	0.250

**Table 2 plants-14-03013-t002:** The K-S test of genome size of induced offspring and self-fertilizing offspring by RY18.

	Sample Size	Average	Standard Deviation	*p*
Induce offspring	51	947.53	54.89	0.092
RY18-F_2_	96	1000.20	81.57	0.200
RY18-F_3_	67	998.14	59.14	0.200

Note: *p* > 0.05 indicates progressive significance and follows a normal distribution.

**Table 3 plants-14-03013-t003:** The analysis of genetic distance.

	RY18	Hybrid F_1_	F_1_ Male Parent
RY18	0	0.1685	0.2304
0068A (common female parent)	0.2663	0.1601	0.2715
C2970 (RY18 male parent)	0.2812	0.3879	0.2839
F_1_ male parent	0.2304	0.2119	0

**Table 4 plants-14-03013-t004:** The correlation analysis of genome size and genetic distance.

	The Genetic Distance Between Parents	Genetic Distance Between Offspring and the Female Parent	Genetic Distance Between Offspring and the Male Parent
The size of the offspring genome	−0.461	0.588	0.074
The size of the male parent	−0.014	0.076	0.790 *
An increase in the size of the genome compared to the male parent’s	−0.475	0.517	−0.621

Note: The Pearson two-tailed test shows a significance *p*-value < 0.05, showing a significant relationship, which is indicated by *.

**Table 5 plants-14-03013-t005:** The correlation analysis between genome size, genetic distance, and over-standard heterosis agronomic traits.

	The Size of the Offspring Genome	The Size of the Male Parent	An Increase in the Size of the Genome Compared to the Male Parent’s	Genetic Distance Between Parents
Plant height/cm	0.041	0.105	−0.033	0.148
Rootstock diameter/cm	0.063	0.025	0.019	0.346 *
Branch height/cm	−0.033	−0.042	−0.001	0.210
Fruiting length of main branch/cm	0.152	0.031	0.106	0.092
Number of primary effective branch/n	0.05	0.012	0.033	−0.021
Number of effective siliques on the entire plant/n	0.175	−0.014	0.147	0.108
Number of effective siliques on the main axis/n	0.195	0.002	0.159	0.149
Number of kernels/g	0.062	−0.077	0.094	−0.097
Thousand-seed weight/g	0.108	−0.310 *	0.300 *	−0.325 *
Theoretical yield per plant/g	0.198	−0.074	0.207	0.033
Actual yield of the plant/g	0.247	−0.166	0.326 *	−0.147
Oil content/%	0.049	−0.099	0.106	−0.087
Glucosinolate/%	0.021	−0.108	0.091	−0.239
Acid/%	0.193	0.072	0.118	−0.260
Protein/%	−0.099	−0.098	−0.013	−0.034

Note: The two-tailed Pearson test shows a significance level of *p* < 0.05, indicating a significant correlation (*).

**Table 6 plants-14-03013-t006:** The information on primer sequences.

Primer Name	Primer Sequence (5’-3’)	Notes
OPSNP7-F	TATATGGGCTGTGCAACGACAAG	*Pol* restoring gene Rf primer
OPSNP7-R	GAGAGAGAGGCTACAGAACAAACT	
Actin-F	TGCTCTTCCTCACGCTATCCTC	actin
Actin-R	GCTCGTAGTTCTTCTCCACCG	

## Data Availability

Data are contained within the article.

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
