# Peer review of "Genome Size Variation Is Associated with Hybrid Vigor in Near-Isogenic Backgrounds in *Brassica napus"

_plants, 2025, doi:10.3390/plants14193013_

Round 1
Reviewer 1 Report
Comments and Suggestions for Authors
This is an interesting study on heterosis and maternal inheritance of DNA markers.
As they identified SNPs in the progeny, it would be of interest to describe where in the DNA those insertions, deletions of bases substitutions took place. This description would help to understand the agronomic traits observed.
The PDF I am sending contains an important number of suggested modifications to improve the undestanding of such findings.
Also, the authors should consider the maternal inheritance in seeds, as observed in cotyledon and testa.

Please review my notes in the PDF
Reviewer 2 Report
Comments and Suggestions for Authors
Dear authors,
Thank you for the opportunity to review your manuscript on genome size variation and heterosis in Brassica napus. This is a creative and technically impressive study that addresses a fundamental question in plant genetics. Your innovative use of the DH inducer line Y3380 to generate progeny with different genome sizes while minimizing genetic distance is a real strength. By combining cytogenetics, flow cytometry, SNP genotyping, resequencing, and agronomic trait evaluation, you provide a rich dataset that opens new perspectives for understanding and applying heterosis in rapeseed breeding.
We find your work novel and promising. At the same time, some revisions would make your conclusions clearer, your data presentation more transparent, and your manuscript more accessible to a broad readership. The following suggestions are offered in the spirit of collaboration, with the aim of helping your study achieve its full potential.
Major Suggestions
-
Please, revise the terminology and Interpretation: The term isogenic may overstate the genetic uniformity of your material. Since measurable genetic distances remain, we suggest using near-isogenic or minimized genetic distance background. Similarly, phrases like “drives hybrid vigor” imply causation, whereas your data show strong correlations. Replacing these with “is associated with” or “may contribute to” in the title and text will make your conclusions more precise and credible.
-
Please, revise the statistical transparency: Report sample sizes and exact p-values for all correlations rather than symbols. Given the number of traits tested, applying or at least discussing multiple testing correction (e.g., FDR or Bonferroni) would reduce the risk of false positives and strengthen your conclusions.
-
Please, take into account the Biological Context and Mechanisms: Expanding your Discussion to include potential mechanisms would enrich the manuscript. For example: nucleotypic effects on cell size/division, gene dosage effects, or repetitive DNA dynamics. If possible, TE content could be explored using your resequencing data. If not, acknowledging this as a limitation and citing relevant literature would still add depth. It would also help to consider whether SNP-enriched regions (A06, C04, etc.) overlap with known yield QTLs in B. napus.
-
Please, revise the Clarity of Experimental Design: Your workflow is complex and currently described across several sections. A schematic figure illustrating the full pipeline—from Y3380 × RY18 cross, through DH induction and selection, to hybrid evaluation—would make the design much clearer for readers.
-
Please, take into account the Scope and Limitations: Since the study is based on progeny from a single hybrid and one environment, please acknowledge these limitations directly and suggest how future work could test generality across additional hybrids and environments.
Minor Suggestions: Define “over-standard heterosis” clearly in the Methods. Refine the SNP analysis description with more technical detail (beyond Excel-based filtering). Simplify data tables (e.g., avoid redundant columns in Table A3). Streamline the Introduction to emphasize the research gap more directly. Ensure references are complete and consistently formatted.
With regards,
Comments on the Quality of English LanguageDear Authors,
Thank you for the opportunity to review your interesting manuscript on genome size variation and heterosis in Brassica napus. I found the scientific content valuable and the experimental approach innovative. Your use of the DH inducer line Y3380 to generate progeny with variable genome sizes while minimizing genetic distance is particularly clever—it allows you to isolate a variable that's typically entangled with many others in heterosis research.
The English language in your manuscript is generally understandable, allowing the scientific content to come through clearly. However, there are several specific areas where improvements would enhance the professionalism, clarity, and precision of your work. Below I've highlighted concrete examples that you should particularly focus on during revision.
1. Please, take care with the Overuse of Causal Language: Your manuscript frequently uses strong causal language that isn't fully supported by your correlational data. This is especially important because your experimental design demonstrates associations rather than proving causation. Specific examples to revise: Title: "Genome Size Variation Drives Hybrid Vigor in Isogenic Backgrounds" Suggestion: "Genome Size Variation is Associated With Hybrid Vigor in Near-Isogenic Backgrounds". Abstract: "correlations with heterosis over the standard for both TSW(r=0.300*) and plot yield(r=0.326*)" Suggestion: "significant correlations were observed between paternal genome size and heterosis over the standard for both TSW (r = 0.300, p < 0.05) and plot yield (r = 0.326, p < 0.05)". Results section (line 232): "It further indicates that there is a significant correlation between the size of the male parent genome and heterosis." Suggestion: "These results indicate a significant correlation between the size of the male parent genome and heterosis." Conclusion (line 26): "Utilizing the doubled haploid (DH) induction line allows for the generation of parental lines with distinct genome sizes, offering a novel strategy for exploiting heterosis in Brassica napus." Suggestion: "The doubled haploid (DH) induction line enables the generation of parental lines with distinct genome sizes, potentially providing a novel approach for heterosis research in Brassica napus." Moreover, using "drives," "allows for," or "offering" implies causation that your correlational data cannot support. This overstatement could undermine the credibility of your otherwise strong findings.
2. Please, revise the Terminology Precision: Your manuscript often uses "isogenic" to describe the genetic background. However, true isogenic lines are genetically identical, while your data (Table 1) show measurable genetic distances between induced offspring and RY18 (average = 0.201). Specific examples to revise: Title: "Genome Size Variation Drives Hybrid Vigor in Isogenic Backgrounds" Suggestion: "Genome Size Variation is Associated With Hybrid Vigor in Near-Isogenic Backgrounds". Abstract (line 13): "whether parental genome size variation alone can generate heterosis under isogenic backgrounds remains unexplored" Suggestion: "whether parental genome size variation alone can generate heterosis under near-isogenic backgrounds remains unexplored". Throughout the manuscript: Replace "isogenic" with "near-isogenic" or "minimized genetic distance background" where appropriate. Therefore, using precise terminology enhances scientific accuracy and helps readers understand the actual scope and limitations of your experimental design.
3. Please, I suggest to revise Statistical Reporting and Figure Legends: Several figures and tables lack essential details that would help readers interpret your results accurately.
Specific examples to revise: Figure 2: "The ploidy identification and chromosome behavior of induced progeny.(A~D) The flow histograms of 157-16, 3689-4, 3689-14, 3689-26;(E~H) The mitosis of 157-16, 3689-4, 3689-14, 3689-26;" Issue: The knowledge base mentions "chromosomal lag" for 3689-26 but the figure legend doesn't explicitly explain what the arrows indicate. Suggestion: Add "Arrows indicate lagging chromosomes in 3689-26" to the legend. Figure 4: "The histogram of genome size of induced offspring and self-crossing progeny by RY18." Issue: Error bars are present but not defined. Suggestion: "Histogram of genome size of induced offspring and self-crossing progeny by RY18. Error bars represent mean ± standard deviation (n = 3-5 plants per line)." Table A4 and Table 5: Contains asterisks (*) denoting significance without definition of what they represent or the sample size (n). Suggestion: Replace with exact p-values and sample sizes (e.g., r = 0.326, p = 0.032, n = 30). Also, consider applying a correction for multiple testing (e.g., Bonferroni or FDR) given that you're analyzing multiple traits.
Why this matters: Without these details, readers cannot properly evaluate the reliability and significance of your results, which is essential for reproducibility.
4. Methods Section Language
The methods section contains several instances of informal language that would benefit from more technical phrasing.
Specific examples to revise:
-
Section 4.9.3 (lines 412-413): "Use Microsoft office excel to identify the common and differential site of high and low productivity, and use the R language CMplot package to plot the distribution of these site on the chromosome." Suggestion: "High- and low-yield-associated SNPs were identified by comparing allele frequencies between high-yielding (top 25%) and low-yielding (bottom 25%) F₁ hybrids. Common SNPs present in both groups were excluded to isolate yield-specific variants. The chromosomal distribution of these SNPs was visualized using the CMplot package in R."
-
Section 4.3 (lines 336-338): "The process of restoring gene primer PCR includes 30 cycles of initial denaturation at 94°C for 3 minutes, denaturation at 94°C for 30 seconds, annealing at 57°C for 30 seconds, extension at 72°C for 1 minute, and final extension at 72°C for 5 minutes." Suggestion: "PCR amplification for restorer gene identification was performed using the following thermal cycling conditions: initial denaturation at 94°C for 3 minutes; 30 cycles of denaturation at 94°C for 30 seconds, annealing at 57°C for 30 seconds, and extension at 72°C for 1 minute; followed by a final extension at 72°C for 5 minutes."
Why this matters: Methods sections require precise, technical language to ensure reproducibility. Informal phrasing like "use Microsoft office excel" undermines the scientific rigor of your methodology.
5. Sentence Structure and Flow
Some sentences are overly long or contain grammatical issues that affect clarity.
Specific examples to revise:
-
Line 401-402: "three low-yield F1(F1-2, F1-3 and F1-13) were selected, and these seven F1 were combined with the control RY 18 and their respective parent samples." Suggestion: "Three low-yield F1 hybrids (F1-2, F1-3, and F1-13) were selected. These seven F1 hybrids were then combined with the control RY18 and their respective parent samples."
-
Line 218-219: "Hybrid offspring were generated by crossing the sterile line 0068A(female parent) with the induced offspring and the self-crossing progeny of RY18, respectively." Suggestion: "Hybrid offspring were generated by crossing the sterile line 0068A (female parent) with the induced offspring. The self-crossing progeny of RY18 served as a control."
Why this matters: Clear, grammatically correct sentences are essential for communicating complex scientific concepts effectively to an international audience.
Reviewer 3 Report
Comments and Suggestions for Authors
The manuscript explores the effects of genome size variation on hybrid vigor in an isogenic background in rapeseed. However, the exact causes of the heterosis effects are still not clear in general and in rapeseed in particular (also discussed in the introduction). Therefore, the effects identified in the paper could not be considered the main drivers of the observed hybrid vigor, but rather one of the contributing factors. Furthermore, the manuscript deals with a single crop species, while the title is very generalistic - giving the misrepresentation that a general driver for all living organisms is provided in the text.
The main plea in the title (that the genome size variation is THE driver of the heterosis effect) is also unsubstantiated. This is due to the well-documented high variability of c-values in many species (even in humans - Nature. 2023 Aug 23;621(7978):355–364. doi: 10.1038/s41586-023-06425-6) which could not be assigned as drivers of hetherosis effects. Consequently, I would propose modifying the title to "Genome size variation affects hybrid vigor in rapeseed isogenic background".
Some incorrect statements should be eliminated and the related conclusions modified. An example is the statement in Lines 166-167 that "all three populations—induced offspring, RY18-F2, and RY18-F3— followed normal distributions (Table 2; Figure 4)". It is apparent (from Fig. 4) that the distribution of "induced offspring" is skewed. My rough estimate is that it is moderately negatively skewed (to the left) and a more thorough statistical analysis of the original data will probably prove it.
The quality of some of the figures should be significantly improved. More specifically:
- Panels A-D of Figure 2
- Figure 4 (including the replacement of Chinese text in axes titles and chart title in the first panel).
The text requires major editing by a professional/native English speaker, as it is difficult to read in its current form due to poor translation and the resulting use of strange terms (i.e., "over-standard heterosis") throughout the text.
Round 2
Reviewer 2 Report
Comments and Suggestions for Authors
Dear authors,
Thank you for the opportunity to review the revised version of your manuscript on genome size variation and heterosis in Brassica napus. I've read through your revisions with genuine appreciation for the thoughtful and thorough way you've addressed the previous review comments. It's always a pleasure to see authors respond so constructively to feedback, and your revisions have significantly strengthened an already interesting study.
With regards,
